# The Evolution of Cooperation in Two-Dimensional Mobile Populations with Random and Strategic Dispersal

Kyle Weishaar [1] and Igor V. Erovenko [2,*]

1    Department of Applied Mathematics, University of Colorado Boulder, Boulder, CO 80309, USA;
     kyle.weishaar@colorado.edu
2    Department of Mathematics and Statistics, University of North Carolina at Greensboro,
     Greensboro, NC 27402, USA
*    Correspondence: igor@uncg.edu

**Abstract:** We investigate the effect of the environment dimensionality and different dispersal strategies on the evolution of cooperation in a finite structured population of mobile individuals. We consider a population consisting of cooperators and free-riders residing on a two-dimensional lattice with periodic boundaries. Individuals explore the environment according to one of the four dispersal strategies and interact with each other via a public goods game. The population evolves according to a birth–death–birth process with the fitness of the individuals deriving from the game-induced payouts. We found that the outcomes of the strategic dispersal strategies in the two-dimensional setting are identical to the outcomes in the one-dimensional setting. The random dispersal strategy, not surprisingly, resulted in the worst outcome for cooperators.

**Keywords:** evolution of cooperation; public goods game; structured population





## 1. Introduction

Understanding the mechanisms that maintain cooperation is a long-standing problem in evolutionary biology [1]. Cooperators sacrifice resources in order to benefit their competitors, suggesting that evolution will actively select against cooperation. Game theory demonstrates [2] that cooperation can be exploited by non-cooperators, also known as free riders or defectors. A free rider is an individual who is assisted by cooperators without sacrificing resources needed to reciprocate any gained benefits. Despite this, natural selection has produced widespread occurrences of cooperation.

Evolutionary game theory has been a useful theoretical framework for identifying mechanisms that encourage cooperation [3–6]. Pairwise interactions between individuals have been described with classical models such as the prisoner's dilemma [2], the snowdrift game [7], and the stag hunt game [8]. Natural interactions can involve multiple individuals. Consequently, multiplayer games [9–11], such as the public goods game [12–14], have been adapted to study cooperation with group dynamics.

Early models [3,15] have demonstrated that infinite well-mixed populations result in a breakdown of cooperation. However, real populations are often divided by spatial or social structures causing individuals to interact primarily with their neighbors. Studies have found the spatial structure to be a crucial mechanism in the evolution of cooperation as it allows cooperators to limit the effects of free riders by forming isolated clusters [6,16–22]. The effect of spatial structure may be dependent on dimensionality as higher dimensions have been shown to be more beneficial to cooperators [23].

Organisms usually have the ability to change their neighbors by traveling to different locations. To account for this, models have introduced mobility in a variety of forms: random dispersal [24–29], moving away from unfavorable locations [30–33], and success-driven mobility where individuals tend towards locations with greater benefits [34–37].

Mobility can assist cooperators by enabling them to form protective groups; however, it can also hinder cooperation by allowing free riders to invade and exploit such groups [38].

In this paper, we study the effects of higher dimensionality on the models established in [35,36]. We consider a finite fixed-sized population of cooperators and free riders. The population evolves through a birth–death Moran process [39] where reproduction depends on the outcome of a multiplayer game [40]. Each individual occupies a node in a square lattice with periodic boundaries (i.e., a two-dimensional torus). Mobility is incorporated via random dispersal and three methods of strategic (success-driven) mobility: probabilistic [36], semi-deterministic [35], and deterministic [35] dispersal strategies.

## 2. The Model

We adopted the one-dimensional models established in [35,36] and expanded them to the two-dimensional spatial structure. We considered a fixed sized population of $N$ individuals, who employed a lifetime strategy of being either a cooperator or a free rider. We used the standard assumption of evolutionary graph theory that mutation occurs on a much slower time scale than selection; for the effect of the variable mutation rate, see [41]. Each individual occupied a node within a 40 by 40 lattice with periodic boundaries. Individuals had the ability to move around the lattice, and multiple individuals could share a single node.

Evolution of this finite structured population was simulated by a stochastic model based on a Markov process. The process was comprised of two types of events: reproduction and dispersal. The reproduction and dispersal processes take place on different spatial scales, as will be explained below.

Every reproduction event begins with each individual playing a public goods game with all individuals that belong to their *interaction neighborhood.* It is a Moore neighborhood of radius $D$, which is one of the parameters of the model; this neighborhood consists of $(2D + 1)^2$ nodes. The payout of the public goods game to the individual $I_n$, whose group consists of $c_n$ cooperators and $f_n$ free-riders (including the individual), is given by

$$p_n = \frac{1}{c_n + f_n} + B \cdot c_n - C_n, \tag{1}$$

where

$$C_n = \begin{cases} C & \text{if } I_n \text{ is a cooperator} \\ 0 & \text{if } I_n \text{ is a free-rider,} \end{cases} \tag{2}$$

and $B$ represents the benefits of cooperation, while $C$ represents the cost of cooperation. Only the cooperators produce the benefit $B$ and pay the cost of cooperation $C$, but all individuals enjoy the benefits provided by cooperators. The term $\frac{1}{c_n + f_n}$ reflects local competition for equally shared standing resources. Free-riders increase local competition without adding benefit to the overall payout; consequently, their presence lowers the payout to every individual in a neighborhood. The benefits provided by cooperators are additive. Biologically, this can be justified as sharing valuable information such as predator proximity [42] or forming biofilms where bacteria can secrete nutrients that are absorbed by their local community [43].

The reproduction events follow the standard invasion process from evolutionary graph theory. This is a BDB (birth–death–birth) Moran process that ensures constant population size [44]. The payout to each individual obtained from one round of the public goods game is converted to the individual's fitness via a smoothing function

$$r_n = \tan^{-1}(p_n) + \frac{\pi}{2}. \tag{3}$$

This function ensures that individuals with a negative payout, as well as those outside large clusters of cooperators, will still have a reasonable chance to reproduce. The probabil-

ity that the individual $I_n$ is chosen for reproduction is directly proportional to their fitness; it is given by

$$\frac{r_n}{\sum_{k=1}^{N} r_k}. \tag{4}$$

Each individual reproduces asexually by producing a single offspring. The offspring will inherit the parent's strategy (cooperator or free rider) and is placed at a random node within the parent's strategic interaction neighborhood. The reproduction event ends with a random (with uniform probability) individual (except for the newborn) being removed from the population.

At every dispersal event, a single individual is randomly (with uniform probability) chosen to move to a location within their *dispersal neighborhood*. The dispersal neighborhood of an individual is a Moore neighborhood of radius $R$, which represents the dispersal range; it is one of the model parameters. The individual "samples" each of the $(2R + 1)^2$ locations within their dispersal range and chooses which location to move to according to one of the following four strategies:

1. Random dispersal: An individual randomly (with uniform probability) selects any location from their dispersal neighborhood.
2. Probabilistic dispersal [36]: An individual will move from their current location with payout $p_n$ to a location that will result in payout $p'_n$, with a probability proportional to $\exp(p'_n - p_n)$.
3. Almost Deterministic dispersal [35]: An individual moves to the location that offers the highest payout; if the highest payout occurs at multiple locations within the dispersal neighborhood, then the individual will randomly choose one of these locations.
4. Deterministic dispersal [35]: An individual will move to the closest location among those that offer the highest payout; if several locations satisfy this criterion, then the individual will randomly choose one of these locations.

We allowed only one dispersal strategy at a time; all individuals adopted the same strategy. This resulted in four different exploration scenarios: one random and three strategic.

The model was initiated by randomly (with uniform probability) placing every individual onto a lattice node. Additionally, individuals were randomly assigned (with uniform probability) the role of either cooperator or free rider. The Markov process proceeded in discrete time steps. The type of the next event—reproduction or dispersal—was determined randomly. The probability of the dispersal event occurring was equal to $\frac{M}{M+1}$, where $M$ is a model parameter representing the population mobility rate.

We created an exact stochastic simulation of this Markov process in Matlab and ran the simulations on an HPC cluster. Each process was run until it reached an absorbing state: the population consisted of only cooperators or only free riders. Each combination of parameters was run 10,000 times, and the fraction of trials that resulted in the population consisting of all cooperators was the fixation probability of cooperators. The accuracy of the simulated fixation probability of cooperators was estimated using the binomial distribution. The standard deviation was equal to $\sqrt{q(q-1)/n}$, where $n$ is the number of trials, and $q$ is the actual fixation probability of cooperators. The maximum value of this expression occurred when $q = 0.5$. Therefore, if $n$ = 10,000 then the standard deviation would be at most 0.005. It follows that our simulated fixation probabilities were correct up to $\pm 0.01$ with a 95% confidence interval.

All model parameters are summarized in Table 1. We used the same range of values for $B$, $C$, $M$, and $N$ as in [35,36]; these references also provide detailed justifications for the choice of these values. The values of $L$, $D$, and $R$ were adapted to the two-dimensional lattice.

**Table 1.** Summary of the model parameters

| Parameter | Meaning | Range of Values |
|---|---|---|
| $B$ | Benefits of cooperation | $\{C/2, C/2 + 1/4, C/2 + 1/2, C/2 + 3/4\}$ |
| $C$ | Cost of cooperation | $\{2, 3, 4\}$ |
| $D$ | Neighborhood radius | $\{1, 3, 6, 10, 15\}$ |
| $L$ | Lattice size | $40 \times 40$ |
| $M$ | Mobility rate | $\{0, 1, 10\}$ |
| $N$ | Population size | $\{10, 20, 30, 40\}$ |
| $R$ | Dispersal range | $\{1, 2, 3\}$ |

## 3. Outcomes for Static Populations

We studied the effects of the model parameters that were independent of mobility, such as spatial structure, by considering an immobile population. We set the mobility rate to zero, which ensured that only the reproduction event occurred. We observed that the general behavior of a two-dimensional static population was consistent with that of the previous one-dimensional models [35,36].

Cooperators thrived in small strategic interaction neighborhoods, while large neighborhoods caused a decline in their performance. This is due to large neighborhoods limiting the number of independent clusters that can exist. For example, a neighborhood radius of 10 allowed at most 3 separate clusters to form. When only a few clusters were available, the population became essentially well mixed. Cooperators are disadvantaged in a well mixed population, because free riders do not pay the cooperation cost. As a result, the fixation probability in large neighborhoods was under 50% regardless of all other model parameters.

The model utilized a fixed lattice; consequently, the population size was equivalent to population density. We considered a lattice with the constant shape of 40 by 40, offering 1600 occupiable nodes. The largest population we considered was 40 individuals; therefore, at most, 2.5% of the available nodes were filled. Consequently, even relatively dense populations were very sparse.

Cooperators tended to have a high fixation probability in dense populations. Figure 1 shows that the fixation probability was greatly boosted by increasing the population size from 10 to 20. Increasing the population size was especially beneficial in situations where cooperators were unfavored due to the high cost of cooperation. When the cost was at its highest ($C = 4$), every population increase significantly improved the fixation probability. In settings with a low cost of cooperation, increasing the population beyond 30 yielded only marginal benefits for cooperators.

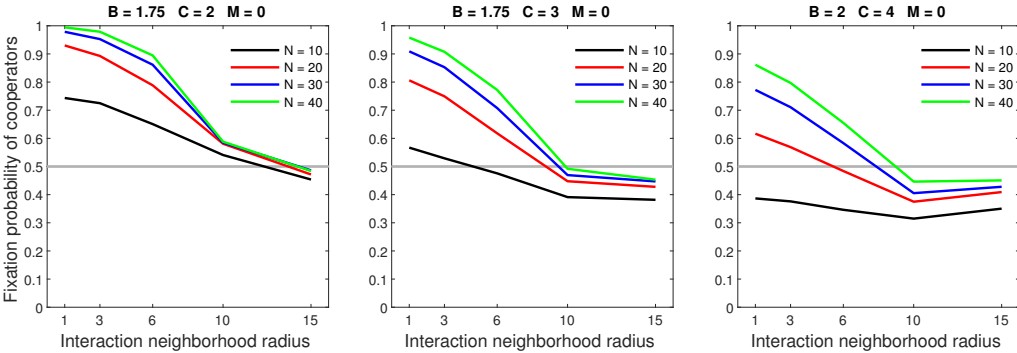

**Figure 1.** Increasing the population density raises the fixation probability of cooperators. The left pane represents the cooperator's ideal condition with the highest benefits of cooperation paired with the lowest cooperation cost. The right pane represents their least favorable condition with the lowest benefits and highest cost.

The survival of cooperators often depends on their ability to form isolated clusters. Consider two separate clusters: one that is completely comprised of cooperators and

another that is infiltrated by free riders. Free riders in the mixed type cluster would have the highest reproduction propensity; thus, they would outcompete their neighboring cooperators. With fewer cooperators present, the reproduction propensities of individuals in the mixed type cluster would decrease. As a result, individuals in the fully cooperative cluster would be more likely to reproduce than those in the exploited cluster. Eventually, the fully cooperative cluster would drive the mixed cluster to extinction.

Population density favored cooperators by increasing the likelihood that an isolated cluster would aggregate as well as limiting the effects of free riders in large clusters. The previous one-dimensional models found that dense populations needed more individuals to be replaced in order to reach an absorbing state and, therefore, offer more reproduction events [35,36]. Consequently, dense populations provide isolated cooperators more opportunities to reproduce and form clusters, whereas sparse populations cause isolated cooperators to be removed before they can aggregate. Dense populations allow clusters to be comprised of a greater number of individuals. In such clusters, the smoothing function limits the differences between the reproduction propensities of free riders and cooperators, thereby reducing the advantage of free riders.

Increasing the benefits of cooperation, while fixing the cost, raised the fixation probability. Figure 2 shows which populations were most favored by increasing benefits. Increased benefits had the greatest effect in sparse populations with a low neighborhood radius. Cooperators in dense populations were heavily dominant; therefore, increased benefits only provided marginal rewards. These behaviors were consistent for every cooperation cost considered.

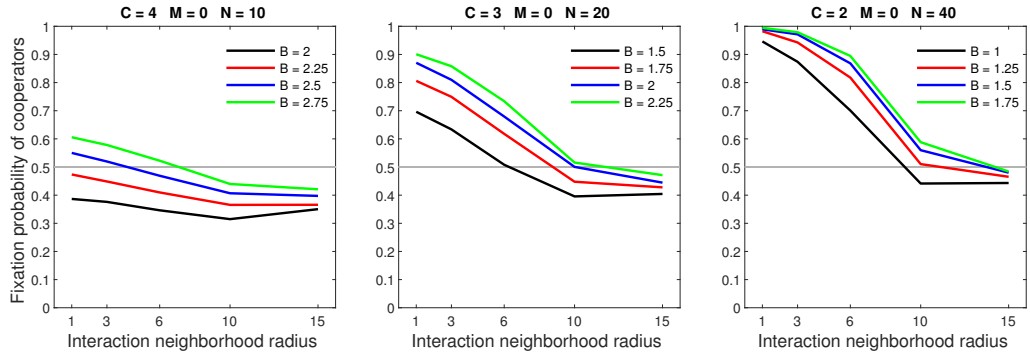

**Figure 2.** Cooperation is promoted by increasing the benefits of cooperation while keeping the cost fixed. The effects of increased benefits are greatest in sparse populations with small interaction neighborhoods.

The benefits of cooperation determined the viability of small cooperative clusters. This was especially influential in sparse populations where most clusters start as an isolated individual reproducing to create a cluster of two. In such clusters, increased benefits significantly raise the reproduction propensity of both individuals resulting in a greater likelihood that the cluster will continue to grow.

Raising the cost of cooperation while maintaining the same relative benefits decreased the fixation probability. Figure 3 shows that increasing the cost was most detrimental to sparse populations with small neighborhoods. An increased cost resulted in marginal penalties to cooperators in dense populations.

The cost of cooperation lowered the reproduction propensity of cooperators, which in turn reduced the likelihood that an isolated cooperator would form a cluster. An increased cost was particularly detrimental to sparse populations where cooperative clusters often originate from an isolated individual. Additionally, the smoothing function limited the effects of increasing the cost on large groups. Consequently, an increasing cost imposed only minor penalties on cooperators in dense populations or populations with large neighborhood radii.

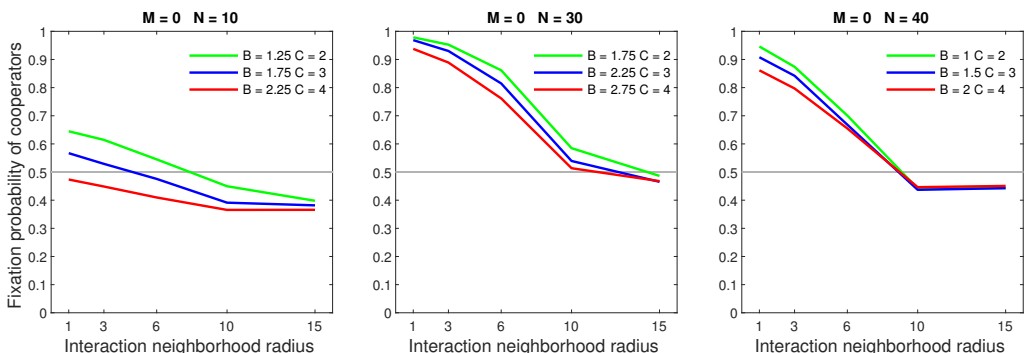

**Figure 3.** Increasing the cost of cooperation while maintaining the same relative benefits decreases the fixation probability of cooperators. The effects are greatest in sparse populations and populations with small interaction neighborhoods.

## 4. Outcomes for Different Dispersal Strategies

Next, we investigated the effect of each dispersal strategy in a two-dimensional setting. The previous one-dimensional models found that high mobility rates and large dispersal ranges tended to lower the fixation probability, specifically in situations where cooperators were dominant [35,36]. These outcomes remained valid in the two-dimensional model.

We studied the effects of the mobility rate by considering an increasing rate while simultaneously fixing the dispersal range at the lowest possible value ($R = 1$). We observed that a low mobility rate generally had limited effects on cooperators. However, a high rate could significantly reduce the performance of cooperators, especially when they were favored by a high population density and small interaction neighborhood radius.

We analysed the impact of increasing the dispersal range while fixing the cost at the central value ($C = 3$). We generally observed that the initial increase in dispersal range (from $R = 1$ to $R = 2$) caused the largest drop in the fixation probability. The subsequent increase in dispersal range (from $R = 2$ to $R = 3$) generally incurred a smaller penalty on cooperators. Increasing the dispersal range was particularly harmful to cooperators in settings with a high population density or a small interaction neighborhood radius.

### 4.1. Random Dispersal

Random dispersal had an overwhelmingly negative effect on cooperators, to the extent that it could lead to a breakdown in cooperation. Figure 4 shows that increasing the mobility rate caused a sharp decline in the fixation probability, especially when the neighborhood radius was 1. An increased rate mildly reduced the performance of cooperators in sparse populations. Low cooperation benefits paired with an increasing rate caused a massive decline in the fixation probability. An increased rate resulted in a similar decline in dense populations.

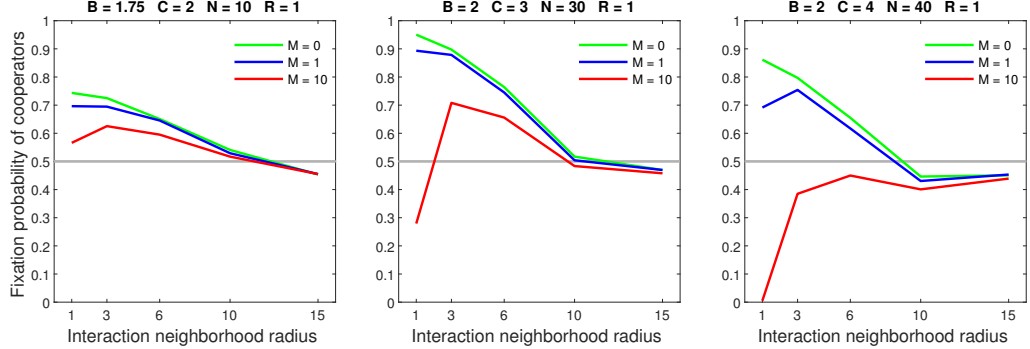

**Figure 4.** In the random dispersal model, increasing the mobility rate is detrimental to cooperators. An increased rate is particularly harmful to cooperators in small neighborhoods with a low benefit–to–cost ratio.

Figure 5 shows that increasing the dispersal range while fixing the mobility rate at 1 could significantly impact cooperation. An increased range had a limited effect on sparse populations. Cooperators with a neighborhood radius of 1 and a low benefit–to–cost ratio were particularly vulnerable to an increasing range. A large dispersal range could cause a breakdown of cooperation in otherwise dominant cooperators. These effects were substantially worst when the mobility rate was 10 (Figure 6). A high mobility rate and large dispersal range always caused a breakdown of cooperation in neighborhoods with radius 1.

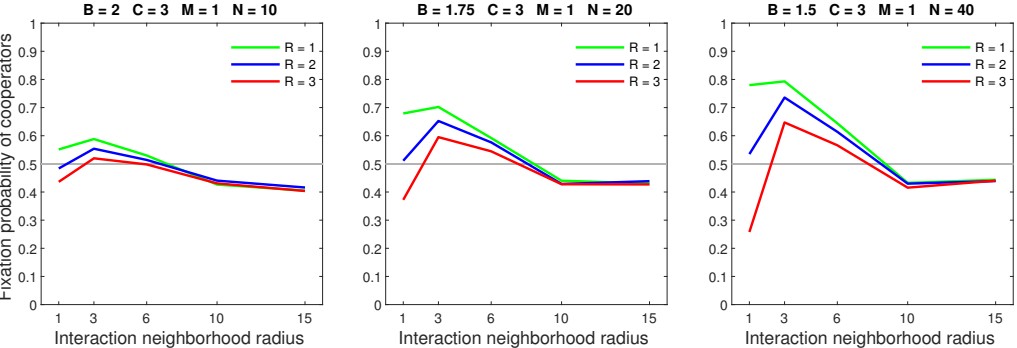

**Figure 5.** In the random dispersal model, increasing the dispersal range for a low mobility rate ($M = 1$) reduces the fixation probability of cooperators in small neighborhoods, especially when $D = 1$.

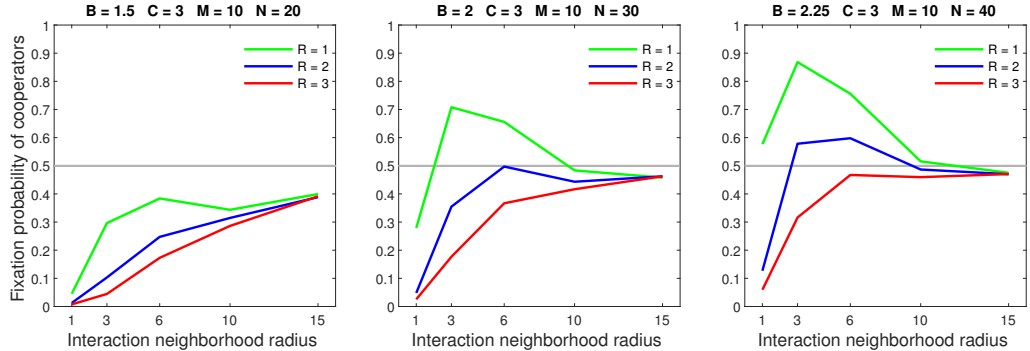

**Figure 6.** In the random dispersal model, increasing the dispersal range for a high mobility rate ($M = 10$) causes a breakdown in cooperation in neighborhoods that are sufficiently small.

Random dispersal was detrimental to cooperators, because it prevented them from forming stable clusters. More specifically, increasing the mobility rate provided clusters more opportunities to drift apart. An increased dispersal range raised the likelihood that an individual would leave a cluster in a single dispersal event. Cooperation can not be maintained when the mobility rate is high and the dispersal range is equal to or greater than the neighborhood radius (i.e., when $D = 1$). In such settings, a breakdown in cooperation is caused by frequently occurring dispersal events having a high likelihood of forcing individuals out of their clusters.

Random dispersal had limited effects on sparse populations where cooperators generally did not have ample time to form clusters. However, increasing the mobility rate and dispersal range disadvantaged cooperators in dense populations, which were heavily dependent on large clusters. Cooperators unable to establish large clusters did not benefit from the smoothing function limiting the effects of free riders. Additionally, dense populations required more reproduction events to reach an absorbing state. Therefore, free riders had more opportunities to benefit from their advantage over scattered cooperators. Cooperators that were unable to form clusters were more sensitive to parameters that affect isolated cooperators. As a result, changes in the cost and benefits of cooperation had a larger effect on the fixation probability.

### 4.2. Probabilistic Dispersal

Probabilistic dispersal hindered cooperation with increases in the movement rate or dispersal range generally leading to a decline in the fixation probability. Figure 7 shows the effects of increasing the mobility rate while fixing the dispersal range at 1. Cooperators already restricted by a sparse population received only light penalties from an increased rate. On the other hand, a high mobility rate reduced the advantage of cooperators in dense populations. Cooperators in neighborhoods with a radius of 1 and a low benefit–to–cost ratio were disproportionately affected by an increased rate.

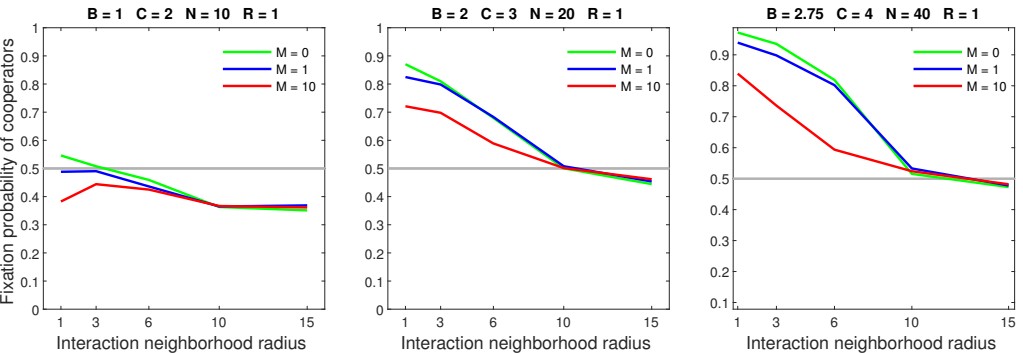

**Figure 7.** In the probabilistic dispersal model, increasing the mobility rate can greatly diminish the advantage of cooperators in dense populations.

Figure 8 shows that increasing the dispersal range while keeping the mobility rate low ($M = 1$) was generally harmful to cooperators. In sparse populations, an increased range primarily affected neighborhoods with a radius of 1. Cooperators in dense populations were broadly undermined by increasing the dispersal range. These effects were greater when the mobility rate was 10 (Figure 9). When the mobility rate and dispersal range were sufficiently large, a breakdown in cooperation occurred in populations with a neighborhood radius of 1 and a low benefit–to–cost ratio.

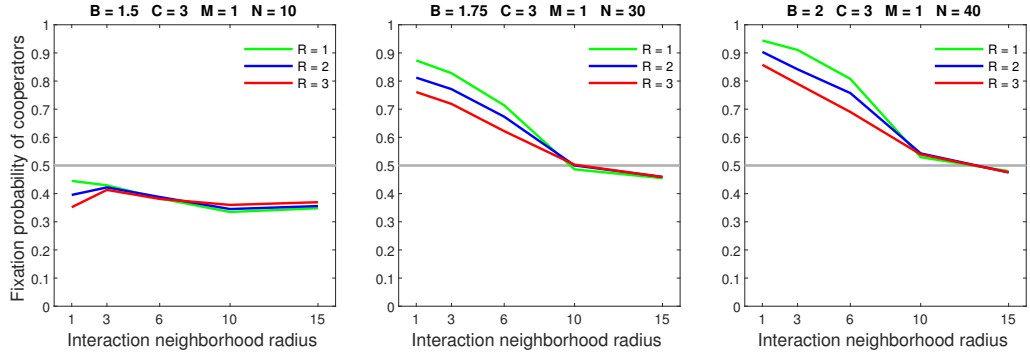

**Figure 8.** In the probabilistic dispersal model, increasing the dispersal range while keeping the mobility rate low ($M = 1$) is primarily detrimental to cooperators in dense populations.

Probabilistic dispersal disadvantaged cooperators in dense populations by allowing free riders to infiltrate cooperative clusters. Increasing the mobility rate offered free riders more opportunities to find and exploit clusters. Sparse populations often led to smaller group sizes and fewer reproduction events. In such settings, free riders had difficulty reaching cooperators in time to exploit them. However, dense populations offered free riders ample time to reach otherwise isolated clusters.

Increasing the dispersal range reduced the number of dispersal events needed for an individual to reach a group of cooperators. Cooperators usually aggregated via reproduction; thus, an increased range primarily allows free riders to infiltrate and exploit clusters. A large dispersal range can compensate for a low mobility rate, thereby reducing the fixation probability.

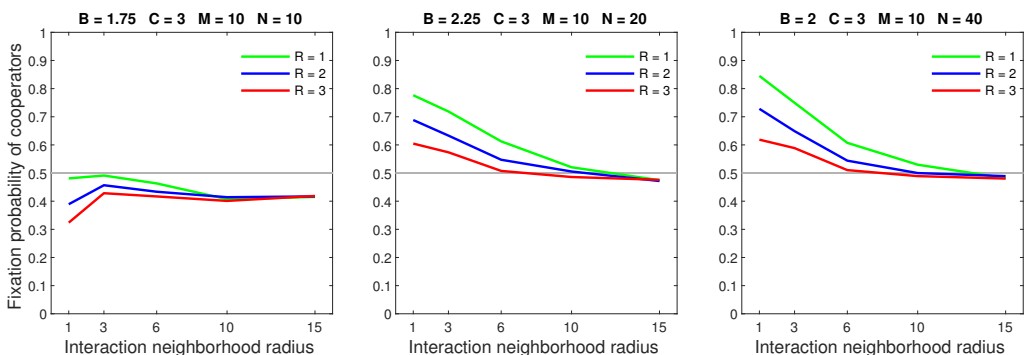

**Figure 9.** In the probabilistic dispersal model, increasing the dispersal range for a high mobility rate ($M = 10$) causes a breakdown of cooperation in sparse populations and eliminates the advantage of cooperator in dense populations.

The observed breakdown in cooperation was the result of a low benefit–to–cost ratio placing heavy penalties on clusters for losing cooperators, exacerbated by a high movement rate and a large dispersal range. A low benefit–to–cost ratio burdened cooperators in small groups. When the dispersal range was larger than the neighborhood radius, there was an elevated chance that cooperators would probabilistically leave their cluster. A high movement rate provides more opportunities for clusters to drift apart. Consequently, the negative impact of the low ratio is enhanced by the average cluster size shrinking.

### 4.3. Almost Deterministic Dispersal

The almost deterministic dispersal was slightly harmful to cooperation with increases in the mobility rate and dispersal range generally having limited effects on the fixation probability. Figure 10 shows that cooperators in dense populations were susceptible to an increasing mobility rate. Sparse populations were widely unaffected by an increased rate, with the expectation of a small decline in the fixation probability of small neighborhoods.

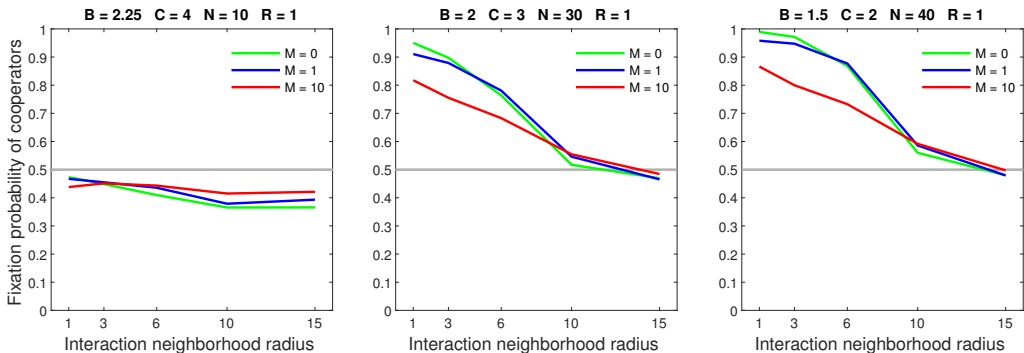

**Figure 10.** In the almost deterministic dispersal model, increasing the mobility rate is detrimental to cooperators in dense populations.

Figure 11 shows the effects of increasing the dispersal range while keeping the mobility rate low ($M = 1$). An increased range reduced the performance of cooperators in dense populations with sufficiently small neighborhoods. These effects were slightly greater when the mobility rate was high (Figure 12). Sparse populations were largely unaffected by changes to the dispersal range, regardless of the mobility rate.

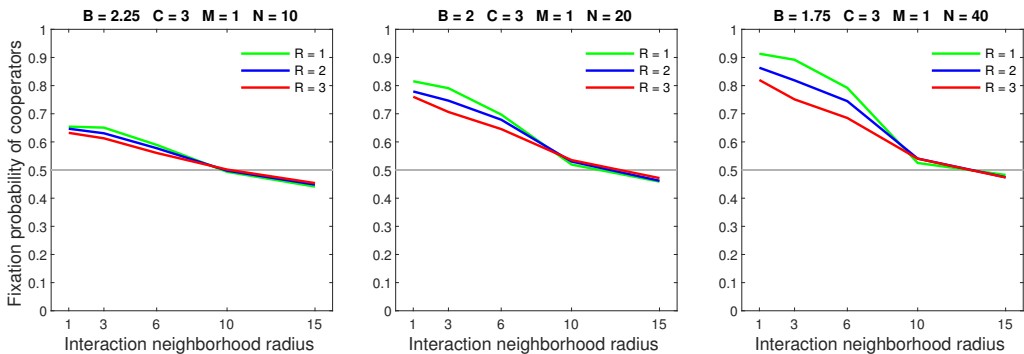

**Figure 11.** In the almost deterministic dispersal model, increasing the dispersal range while keeping the mobility rate low ($M = 1$) is primarily detrimental to cooperators in dense populations.

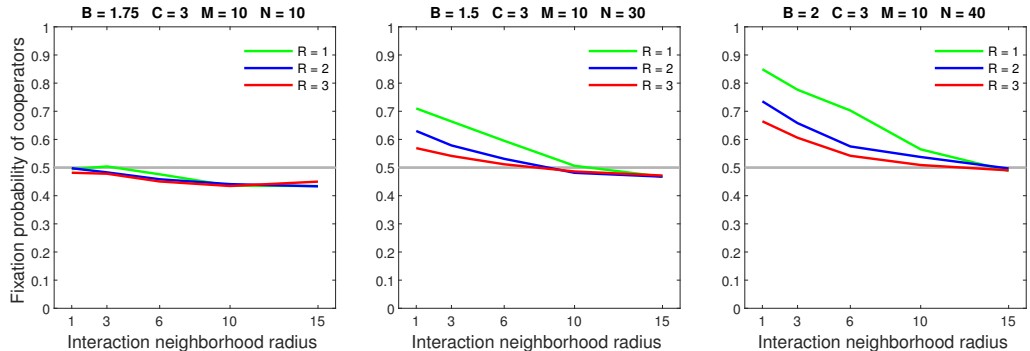

**Figure 12.** In the almost deterministic dispersal model, increasing the dispersal range for a high mobility rate ($M = 10$) affects cooperators in dense populations, while sparse populations remain unchanged.

The almost deterministic dispersal disadvantaged cooperators by allowing free riders to efficiently locate clusters. Individuals using this dispersal strategy always moved to an optimal position. A high mobility rate gave free riders ample opportunities to reach clusters. Similarly, a large dispersal range permitted cooperators to travel farther with each dispersal event. As a result, cooperators in dense populations were unlikely to remain isolated. Sparse populations offered fewer reproduction events and fewer clusters. As such, they did not provide sufficient opportunities for even optimally moving free riders to exploit cooperators.

### 4.4. Deterministic Dispersal

Deterministic dispersal had minor effects on cooperation. Figure 13 shows that increasing the mobility rate had the greatest impact on dense populations. In such settings, even a high rate only caused a small decline in the fixation probability. Cooperators in sparse populations suffered slight penalties from an increased rate.

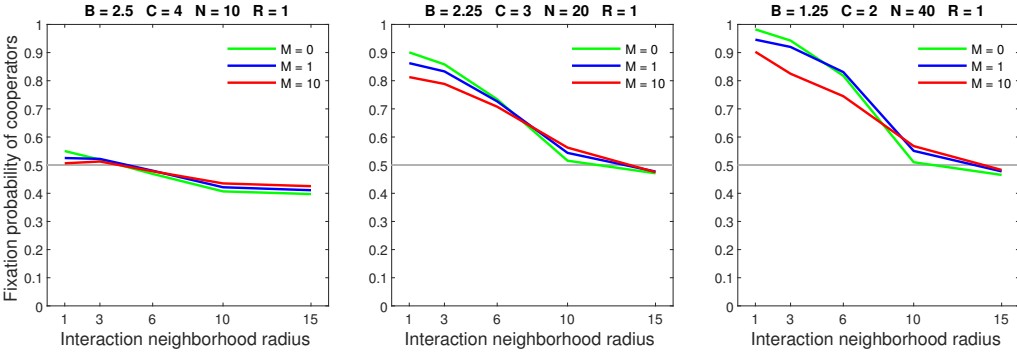

**Figure 13.** In the deterministic dispersal model, increasing the mobility rate primarily lowers the fixation probability of dense populations with small neighborhoods.

Figure 14 shows that increasing the dispersal range while keeping the mobility rate low ($M = 1$) was harmful to cooperators in dense populations. Even when the mobility rate i=was high ($M = 10$), an increased range did not completely eliminate the advantage of dominating cooperators (Figure 15). Regardless of the mobility rate, an increased range had a negligible effect on sparse populations.

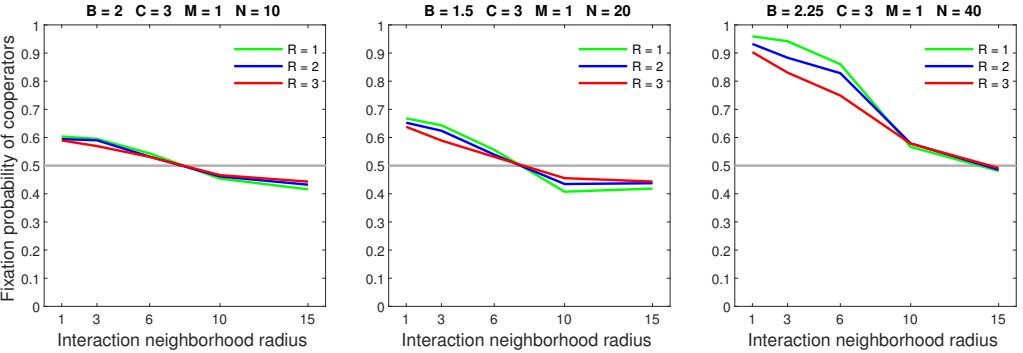

**Figure 14.** In the deterministic dispersal model, increasing the dispersal range for a low mobility rate ($M = 1$) causes small declines in the fixation probability of dense populations while leaving sparse populations unaltered.

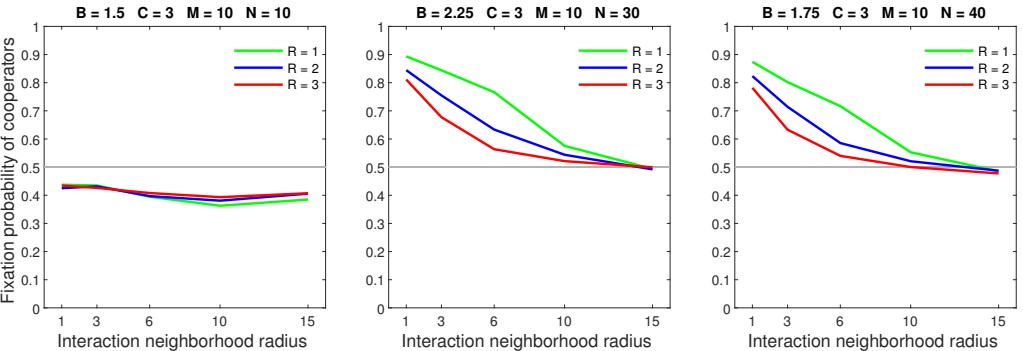

**Figure 15.** In the deterministic dispersal model, increasing the dispersal range for a high mobility rate ($M = 10$) only has a negative effect on cooperators in dense populations.

Deterministic dispersal restricted the movement of free riders; hence, it had a limited effect on cooperation. Free riders must move to the closest optimal position; therefore they could only enter the edge of a cluster. As a result, it was easy for cooperators to flee from encroaching free riders. Increasing the mobility rate and dispersal range primarily affected dense populations, where cooperators formed large clusters. Having too many members prevented a large cluster from traveling faster than oncoming free riders. However, the small clusters seen in sparse populations could effectively retreat from invading free riders.

## 5. Comparison of the Dispersal Strategies

In this section, we compare the outcomes of each dispersal strategy. We observe that the general behavior of each strategy (except random dispersal, which was new for the two-dimensional setting) remained consistent with the results of the one-dimensional models [35,36]. Overall, when the mobility rate was low ($M = 1$), every form of success-driven dispersal produced a similar fixation probability. This is the result of the low rate limiting dispersal opportunities, thereby restricting the impact of mobility. When the dispersal range was small ($R = 1$), random dispersal generally produced a similar fixation probability to the success-driven dispersal strategies (Figure 16), the exception being a reduced fixation probability when the cost of cooperation was high and the interaction neighborhood was small ($D = 1$).

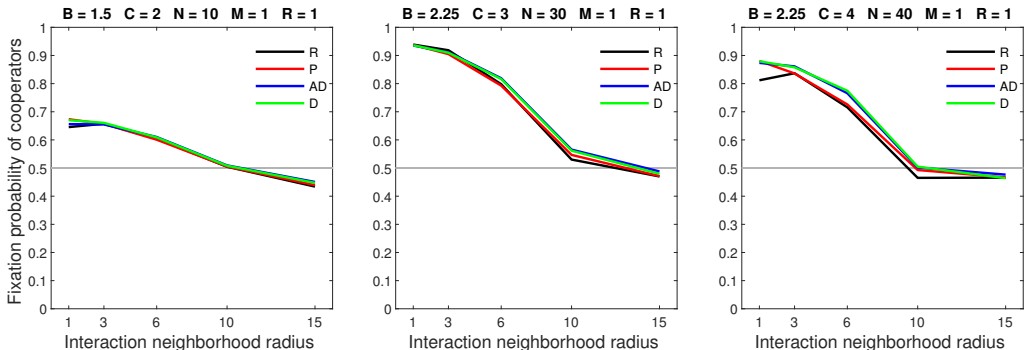

**Figure 16.** All models result in similar fixation probabilities when the mobility rate is low and the dispersal range is small. Legend: R—random dispersal, P—probabilistic dispersal, AD—almost deterministic dispersal, D—deterministic dispersal.

A low mobility rate ($M = 1$) and a medium dispersal range ($R = 2$) demonstrated the advantage of success-driven dispersal models over random dispersal. Random dispersal yielded lower fixation probabilities in settings with sparse populations or low benefits of cooperation (Figure 17).

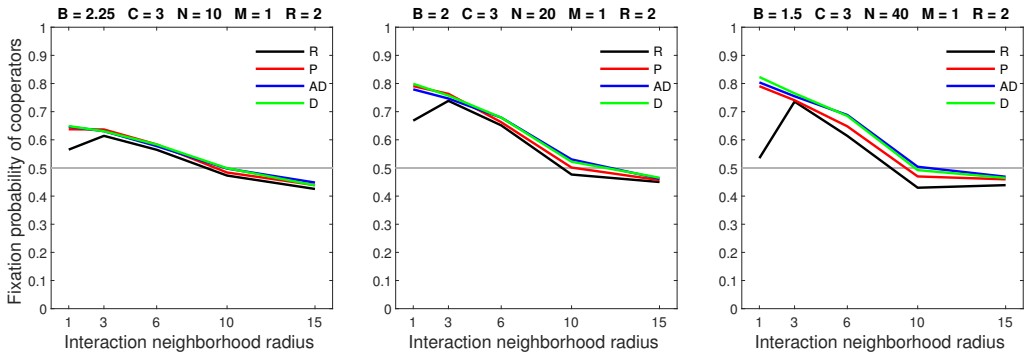

**Figure 17.** When the mobility rate is low and the dispersal range is sufficiently large, all forms of success-driven mobility result in higher fixation probabilities than random dispersal in neighborhoods of radius 1.

A low mobility rate paired with a dispersal range of 3 resulted in differing model outcomes. (Figure 18). The deterministic and almost deterministic dispersal models produced similar fixation probabilities. However, probabilistic dispersal reduced the fixation probability in populations with a low benefit–to–cost ratio and a neighborhood radius of 1. The random dispersal model often resulted in significantly lower fixation probabilities when $D = 1$.

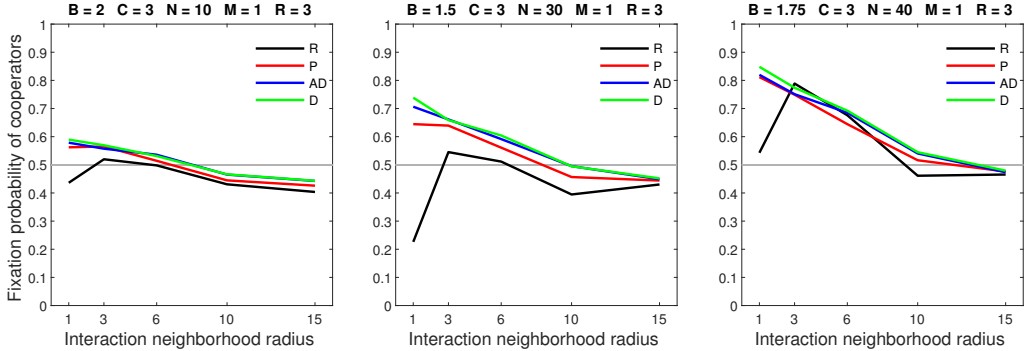

**Figure 18.** Deterministic and almost deterministic dispersal are more favorable to cooperators than probabilistic dispersal in settings with a low benefit–to–cost ratio and a neighborhood radius of 1.

When the mobility rate was high ($M = 10$), the fixation probability primarily depended on the cooperators' ability to evade free riders and maintain stable clusters. Entering a cooperative cluster significantly increased a free rider's payout. However, cooperators moving away from free riders only received a marginal increase in payout resulting from less competition for local standing resources, reflected in the term $\frac{1}{c_n + f_n}$. Consequently, every considered form of success-driven mobility allowed free riders to effectively exploit clusters, while the cooperators' ability to evade free riders varied (Figure 19).

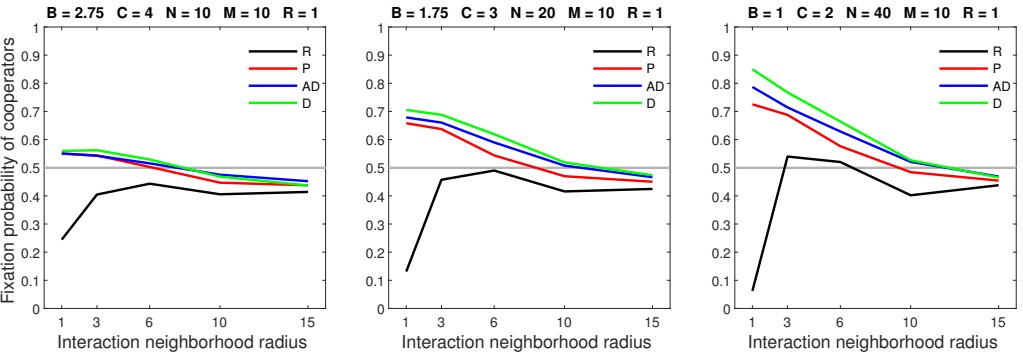

**Figure 19.** Deterministic dispersal is the highest performing model, followed by the almost deterministic model, with probabilistic dispersal producing the lowest fixation probability of out of all success-driven models.

The probabilistic dispersal model was ill suited for evasion, since the slightly higher payout only mildly raised the probability that a cooperator would move away from free riders. However, cooperators using the deterministic or almost deterministic dispersal strategies relocated to a location with the greatest payout, regardless of how much of an improvement it would be. Additionally, deterministic dispersal only allowed free riders to enter the edge of a cluster. As a result, cooperators could flee from free riders with fewer dispersal events.

The random and probabilistic dispersal models allowed clusters to drift apart and were, therefore, less favorable to cooperators. In both models, a low benefit–to–cost ratio led to a breakdown in cooperation (Figure 20). Such breakdowns did not occur with the deterministic and almost deterministic dispersal strategies, since cooperators were able to form stable clusters.

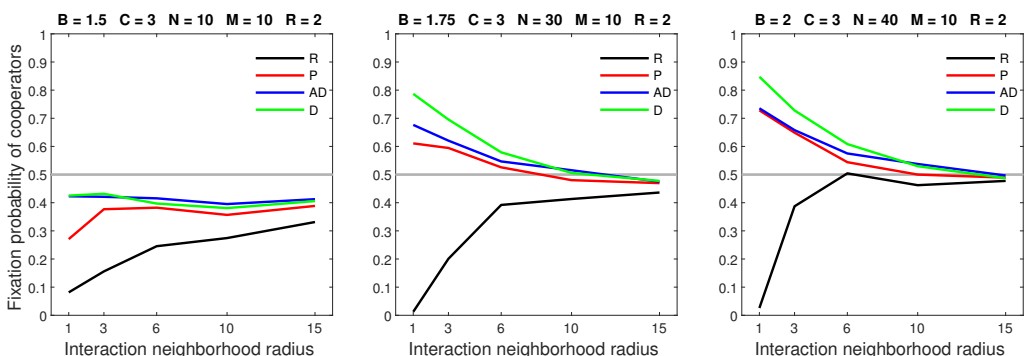

**Figure 20.** Breakdowns in cooperation only occur in the random and probabilistic dispersal models.

A high movement rate ($M = 10$) and large dispersal range ($R = 3$) demonstrated the general ranking of each dispersal strategy (Figure 21). Random dispersal resulted in the lowest fixation probabilities as it prevented cooperators from maintaining clusters. The probabilistic model did not allow cooperators to effectively evade free riders and suffered from clusters drifting apart. Therefore, probabilistic dispersal was the lowest performing form of success-driven mobility. Cooperators using the almost deterministic strategy could avoid encroaching free riders. However, free riders were able to enter the

center of clusters, which made it difficult for cooperators to break away. As such, almost deterministic dispersal was the middle performing form of success-driven mobility. The deterministic model was the highest performing dispersal strategy as it allowed cooperators to evade oncoming free riders. Consequently, deterministic dispersal was the only model that preserved cooperators' dominance in dense populations, even when the mobility rate was high, and the dispersal range was large.

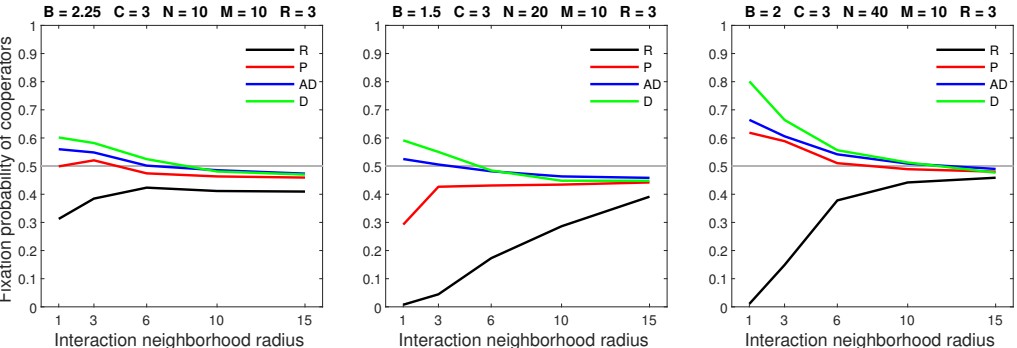

**Figure 21.** The deterministic dispersal is the only model that maintains a cooperator's advantage in dense populations.

## 6. Discussion

This research was motivated by our desire to investigate the effect of the dimensionality of the spatial structure on the models investigated in [35,36]. Compared to the previous one-dimensional models, the two-dimensional model provided cooperators additional directions to escape from encroaching free riders. Additionally, in the two-dimensional model, the area of the dispersal range and strategic interaction neighborhood changed exponentially. Yet, the overall qualitative findings of [35,36] for the one-dimensional model remained valid for the two-dimensional model:

- Cooperators do best in small strategic-interaction neighborhoods and higher population densities.
- Cooperators may achieve fixation probabilities close to one in "good" circumstances.
- Deterministic dispersal—where individuals move to the closest location with maximum payout—avoids a breakdown in cooperation in "bad" circumstances.

We found that the general behavior of each strategic dispersal strategy did not change in the two-dimensional setting. The ranking of the dispersal strategies in terms of the evolution of cooperation were consistent with the one-dimensional results, with deterministic dispersal being the best performing form of success-driven mobility and probabilistic dispersal being the worst.

The results of this paper, therefore, suggest that the model is robust with respect to the lattice dimension, and that the original one-dimensional model already captures all the key features.

One novel feature of the two-dimensional model that was not considered in the one-dimensional models was the random dispersal strategy. The random dispersal strategy is similar to the exploration strategy in the territorial raider model [40]. The random dispersal offers new insights into the breakdown in cooperation observed in the probabilistic model [36]. Mobility can harm cooperation by allowing free riders to infiltrate and exploit cooperative clusters. However, dispersal strategies that prevent stable clusters from forming are severely detrimental to cooperators. Therefore, the observed breakdown in cooperation under random dispersal was the result of cooperative clusters drifting apart.

In our model, movement incurred no cost. However, this is not necessarily realistic; see [45] for the model of a mobile population with costly movement. The cost of movement may have a deciding effect on the evolution of cooperation. For example, in [45], the cost of movement was the main predictor of the stability of the population of defectors.

With the success-driven dispersal strategies, the movement of individuals depends on the locations of the other individuals, because all individuals tend to be in the vicinity of cooperators while avoiding free-riders. In essence, the individuals do not move independently of each other. Yet in our current model, the dispersal of individuals depended on the current locations of other individuals rather than their dispersal strategies. See [46] for a collection of novel models of coordinated movement, which provide various measures of dispersal and aggregation in groups of individuals.

**Author Contributions:** K.W.: running simulations, visualizations, manuscript writing; I.V.E.: study design, coding simulations, manuscript writing, project supervision. All authors have read and agreed to the published version of the manuscript.

**Funding:** This research received no external funding.

**Institutional Review Board Statement:** Not applicable.

**Informed Consent Statement:** Not applicable.

**Data Availability Statement:** Not applicable.

**Conflicts of Interest:** The authors declare no conflicts of interest.

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
