# Peer review of "The Evolution of Cooperation in Two-Dimensional Mobile Populations with Random and Strategic Dispersal"

_games, doi:10.3390/g13030040_

Round 1
Reviewer 1 Report
This paper studies conflicts between cooperators and free riders in a two-dimensional lattice. Individuals are located at the sites of the lattice and play the publica goods games with their neighbors. Individuals can move from one site to another, according to a moving strategy (chosen from a deterministic one to a probabilistic one). In the previous papers [13,15], it is found that high mobility rates and large dispersal ranges lower the fixation probability of cooperators for one-dimensional settings. This current paper shows that the same results as the previous papers hold for two-dimensional settings. Major parameters used in the paper are the same as those in the previous papers, in this sense, this paper is a direct extension of the previous ones. I think that the paper is well written and should interest the readers of Games. I recommend its publication after the authors revise their manuscript according to the comments below.
・The reference number starts with[30]. Please fix the order of references.
・In eq.(1)., I did not understand well why the term 1/(c_n+f_n)is “added”. If benefit provided by cooperators is shared among all individuals, BC_n should be “divided” by c_n+f_n.
・In the manuscript, 2 strategies (cooperators or free riders), are considered. The frequency of cooperators in the population evolves in time according to the Moran process. On the other hand, 4 mechanisms about how to disperse individuals (from a deterministic one to a random one) are also called “strategies”. I was confused about this, since it is not mentioned how individuals choose one of the four possible mechanisms. Are there 8 (2 times 4) strategies in total?If individuals do not choose how to disperse but it is set by the authors, it is not good to call them “strategies”. Please make the strategy space clear.
・Since the authors frequently refer to clusters of cooperators in the two-dimensional lattice, it would be kind if illustrations of some typical patterns in the two-dimensional lattice are shown.
Reviewer 2 Report
In general, this is a very interesting research approach towards cooperation in populations that interact through public goods games and reproduce through fitness-based selection. The paper is well-written, the single points are clearly presented. However, the model itself, the choice of model aspects, the presentation of results in whole, and the implications of the results are all problematic points. To consider this paper for publication, a reasonable major revision is necessary.
Major points:
What is most problematic with this model is the fact that interaction is local but the resulting fitness value is global, namely an average over all individual’s scores (4). Especially with such a sparse occupation share of 2.5% we have often the emergence of separate clusters without much or any contact to other clusters. Why is the reproduction rate then a result of a global average? This is a very problematic point, since it is not realistic that an individuals fitness is impacted by other individuals without any contact. One could argue that global fitness is result of an access to global resources, where higher-scoring individuals have more access to. But then we have a local interaction structure, but a global distribution mechanism. Is that realistic? I think the author(s) should elaborate more on this point.
Furthermore, the absorbing states of only cooperators are not very stable when we have mutation, right? I can imagine that whenever a cooperator of a cooperator cluster changes to a defector/free-rider, there is a very high chance that the cluster is replaced by only free-riders. I think that author(s) should elaborate more on the term “absorbing states”, particularly with respect to evolutionary stability and if mutants can be pushed back. If I understand it right, cooperators can only survive if they form isolated clusters without any free-riders. And then the free-riders get replaced over time because they score worse in their own isolated free-rider clusters, since they are affected by the global fitness value (see my former point).
I find an excess of graphical presentations of simulation results. The paper has !21! of the same figure type, with three subfigures each showing the simulation results for different parameter settings. This paper reads for many pages like a technical report, not a research paper. I would clearly suggest to strongly reduce the number of figures in the main paper to those that have the most relevant results, and remove the other figures to the appendix. In the given format we have the problem that the reader can easily loose track of what results are really important for the bigger story, and what are just strongly expected results (e.g. increasing costs/decreasing benefit is detrimental for cooperators…).
When I look at the Figures 16-21 it strikes me that the dispersal methods P, AD and D are almost always produce the same results, with only a few exceptions. It doesn’t not seem to me, that they really produce significantly different results. This is surprising since these methods were prominently introduced (lines 87-98). I am wondering what we learn from this comparison. It is not convincing to me that we need all three methods. Comparing only one of them with R (random dispersal) would be sufficient to make the point, and reduces the parameter spectrum.
The main result of the paper (and in the discussion) is that the model does not behave significantly from the one-dimensional model: “Yet the overall qualitative findings of [15, 13] for the one-dimensional model remain valid for the two-dimensional model. We found that the general behavior of each strategic dispersal strategy did not change in the two-dimensional setting.” This raises two questions:
-
First of all, why don’t we have the overall results of the one-dimensional study presented here (at least in a nutshell). More concretely, what is this general behavior exactly, that we find in the former studies and here?
-
Secondly, what do we gain from this study considering that we found these effects already in a one-dimensional model? See also the next point.
You write: “The results of this paper therefore suggest that the model is robust with respect to the lattice dimension, and that the original one-dimensional model already captures all the key features.” This is a very low gain considering the great number of different simulation experiments that has been made. As a further result, you also write: “The random dispersal offers new insights on the breakdown of cooperation observed in the probabilistic model. Mobility can harm cooperation by allowing free riders to infiltrate and exploit cooperative clusters.” But this is an effect that would, for example, also produced by allowing for mutation (see one of my former points). In general, any ‘noise’ (defector, free-rider) that enters cooperators clusters would make free-riders infiltrate them. This is obvious to begin with. Therefore, I cannot find this a very novel insight at all.
Minor points:
Line 17: public good -> public goods
Line 47/48: „Individuals have the ability to move around the lattice and multiple individuals may share a single node.“ Does this also imply, that nodes can be empty? (Accordint to lines 137-139, it does) Then there is a good chance that an individual might have no neighbors at all. For that case, we have a zero-division in definition (1).
Line 92/96: Why is rule (3) called “Almost Deterministic” and rule (4) called “Deterministic”, although both have some random elements/
Line 96: It is not clear to me what the criterion “closest location that offers the highest payoff” means. In fact, it involves two criteria, being close AND offer the highest payoff.
Line 138-140: I am wondering what happens with much larger populations? Any information would be nice.
Line 139/140: “Consequently, even relatively dense populations will be very sparse.” Why do you call it dense population at all then? This term is somehow misleading, considering a maximal occupation share of 2.5%.
Reviewer 3 Report
Dear authors,
Thanks for reviewing this article. The article is very interesting in terms of changing the behavior of cooperation actions within a defined environment. Therefore, I consider the article to be specific and beneficial in the given area.
Sincerely,
reviewer
Author Response
We thank the reviewer for taking time to read the manuscript and provide useful feedback.
Round 2
Reviewer 2 Report
The author’s revision improved the manuscript in many ways. I am fine with most of the adjustments and arguments. However, there are two aspects of the model that I would like to see more thoroughly discussed, namely
1. Why local interaction but globally-driven selection?
2. Why zero mutation
I understand that both seem to be standard assumption in Evolutionary Graph Theory, but if they are standard assumptions, then they must be well justified by Evolutionary Graph Theorists. It would be helpful to add the sources and the arguments therein, even in a nutshell. I am an Evolutionary Game Theorist (if you want), and for me these assumptions need at least some explanation, because they do not adequately model many phenomena of biological or cultural evolution. If the authors could elaborate a little bit more on these assumptions, it would clearly improve the paper for readers that come from related field but are not experts in Evolutionary Graph Theory.
To be more detailed, here are my responses to the author’s responses to my earlier points (1 to 7):
With respect to the 1. point you write: “the probability of being chosen for reproduction is directly proportional to the fitness. This requires taking into account the fitness of all individuals.”
Why? The term fitness means how well I adapt to the (generally local) environment. I still think that a local fitness value would be a more realistic model. I understand that a global fitness value is a standard assumption, but it comes from models where a homogeneous population structure (everybody interacts with everybody) is assumes, as found e.g. in the replicator equation. I understand that the authors won’t invent a local fitness value and run all the simulations again. That is not what I expect. But there should be more explanation of why a model with strongly local interaction but global fitness and consequential globally-driven selection is realistic. What are the scenarios where groups interact mostly locally, but when it comes to selection of other strategies, they are globally-driven?
2. point: Why does mutation occur on a much slower time scale? If this is standard in evolutionary graph theory, then there should be a good reason for this assumption. It would be nice to be added to the paper. On another note, if mutation occurs on a much slower time scale, then this could be modeled with a very low mutation rate.
3. point: I still think it would be better readable to move some figures to the appendix, but I leave this up the authors.
4. point: Fine with me.
5. point: Well addressed
6. point: Acknowledged.
7. point: Yes, it also should be considered as a negative result. It is definitely not a null-result, but I think also not a very surprising one, which therefore I can only give a low score for originality/novelty and significance of content.
All the remaining minor points are very well addressed.
Author Response
We would like to thank the reviewer for taking time to carefully read the revised version of the paper. We don't think this manuscript is an appropriate "battlefield" where the standard modeling practices of evolutionary graph theory need to be "defended" or justified from the basic principles. Nevertheless, we acknowledge the reviewer's desire to make the paper more accessible to non-experts in evolutionary graph theory.
We have added some additional explanations and references to the manuscript to address the two main points of concern of the reviewer.
- We expanded one paragraph on page 2 that sets up the evolutionary process by adding that this modeling approach is standard in evolutionary graph theory and by including a reference that describes the evolutionary processes on graphs, in particular, the BDB updating mechanism.
- We added the explanation that the slow mutation rate is a standard assumption in evolutionary graph theory, and we added a reference to the investigation of the effect of variable mutation rate on evolutionary games on graphs to the first paragraph in section 2. There do exist models with various mutation rates in evolutionary graph theory, though the fast mutations rates are rarely considered. However, varying the mutation rate was beyond the scope of the current project.